# Research on Speed Control Methods and Energy-Saving for High-Voltage Transmission Line Inspection Robots along Cable Downhill

Zhiyong Yang [1,2], Xu Liu [1,2], Cheng Ning [1,2], Lanlan Liu [3,4], Wang Tian [1,2], Haoyang Wang [1,2], Daode Zhang [1,2,*], Huaxu Li [3,4], Dehua Zou [3,4] and Jianghua Kuang [3,4]

1  School of Mechanical Engineering, Hubei University of Technology, Wuhan 430068, China; yzy017@126.com (Z.Y.); 15991755143@163.com (X.L.)
2  Hubei Key Laboratory of Modern Manufacturing and Quality Engineering, School of Mechanical Engineering, Hubei University of Technology, Wuhan 430068, China
3  Intelligent Live Working Technology and Equipment (Robots) Hunan Provincial Key Laboratory, State Grid Hunan Electric Power Co., Ltd., Ultra High Voltage Transmission Company, Changsha 420100, China
4  Live Inspection and Intelligent Operation Technology State Grid Corporation Laboratory, State Grid Hunan Electric Power Co., Ltd., Ultra High Voltage Transmission Company, Changsha 420100, China
*  Correspondence: hgzdd@126.com

**Abstract:** To ensure the safe operation of high-voltage transmission line inspection robots during downhill descents without power and extend their range after a single charge, this paper proposes an energy-saving speed control method for the inspection robot's walking wheel motor on downhill slopes by integrating feedback braking and fuzzy PID control. By combining the parameter equation of the overhead catenary line and the structural characteristics of the overhead transmission line, this paper analyzes the relationship between the driving torque of the inspection robot's wheels and the horizontal displacement along the transmission ground wire before and after descending. Based on this analysis, a speed control and energy recovery scheme is developed for the inspection robot, which combines front-wheel feedback braking with rear-wheel regenerative braking. The fuzzy PID method is utilized to adjust the PWM duty cycle to achieve energy-efficient speed control of the inspection robot's rear walking wheels. Additionally, to improve the energy density and specific power of the robot's energy storage unit, a composite power source consisting of lithium batteries and supercapacitors is employed to recover energy from the front walking wheels through feedback braking. The combined simulation results indicate that, compared to fuzzy control and PID control, fuzzy PID control better regulates the robot's speed under varying slopes, wind resistance, and cable roughness. A downhill speed control system for the inspection of the robot's walking wheel motor was designed, and its effectiveness was validated through simulated high-voltage line experiments. The fuzzy PID control was demonstrated to effectively maintain the rear walking wheel speed within the targeted range during downhill descents. When descending along a fixed 30° angle cable, the fuzzy PID control resulted in an increase of 5.28% and 14.26% in the state of charge (SOC) of the supercapacitor compared to PID control and fuzzy control, respectively. Moreover, when descending along fixed angle cables of 10°, 20°, and 30°, as well as a variable angle cable ranging from 30° to 0°, the SOC of the supercapacitor increased by 17.55%, 26.25%, 38.45%, and 31.29%, respectively. This demonstrates the effective absorption of regenerative braking energy during the robot's downhill movement.

**Keywords:** inspection robot; regenerative braking; fuzzy PID control; energy recovery; SOC

## 1. Introduction

The high-voltage transmission line inspection robot is an effective tool for achieving automation in examining high-voltage lines. It serves as a replacement for human



inspectors, enabling the performance of inspection tasks in challenging areas such as high mountains, lakes, virgin forests, and desert wastelands that are otherwise difficult for human access [1–3]. Due to the flexibility of high-voltage transmission cables, the downhill force acting on the inspection robot constantly varies throughout the process. Consequently, the downhill speed control system exhibits a non-linear and time-varying nature. If the robot's downhill speed cannot be effectively controlled, significant safety risks to the inspection operation may arise [4]. In addition, inspection robots are typically powered by energy-limited batteries, which lack effective means of energy replenishment during line inspections. Furthermore, the power consumption of the robot's locomotion mechanism and inspection equipment is significant, resulting in a very limited range of operation after a single charging cycle [5]. Therefore, researching new energy-saving technologies and exploring new power supply methods are of great significance in extending the range of operation for inspection robots after a single charging cycle.

In response to the speed control issue for the walking wheel motor of inspection robots along power transmission lines during downhill descents, traditional speed control methods for DC motors typically involve techniques such as weak magnetic speed control [6], series resistor speed control [7], and voltage adjustment speed control [8] to regulate the downhill speed of the robot. Hu et al. effectively controlled the passive downhill speed of inspection robots by employing energy consumption braking and pulse width modulation techniques [9]. Yang et al. proposed a passive downhill speed control method that combines energy-consumption braking with regenerative braking [10]. The aforementioned two-speed control methods both utilize energy-consumption braking to control the downhill operation of the inspection robot, effectively regulating the speed of the walking wheel motor. Currently, there is relatively limited research on downhill speed control methods specifically for inspection robots. However, significant research achievements have been made in the automotive field regarding downhill speed control. In the automotive field, various algorithms, such as dual-loop control [11–13], neural networks [14,15], and Kalman filtering [16,17], are commonly used to control the running speed of vehicles. Zhang et al. implemented speed control for pure electric buses using a dual-loop control system that incorporates both vehicle speed and current feedback [18]. Chen et al. proposed a three-parameter shifting control strategy for automated manual transmission (AMT) vehicles using neural networks. This strategy aims to improve the smoothness of gear shifting in automobiles [19]. Han et al. proposed an active safety control method for downhill driving, aiming to enhance the safety and fuel economy of vehicles on downhill road sections [20].

To address the energy recovery issue for inspection robots operating along power transmission lines, Yang et al. employed a feedback braking method to convert the gravitational potential energy of the inspection robot during downhill processes into electrical energy and store it in the batteries [10]. Jin et al. developed a vehicle downhill control system based on an electronic stability program designed to assist drivers in controlling the vehicle speed during low-speed downhill driving and ensuring safety. Additionally, the system incorporates regenerative braking to recover energy during downhill descents, thereby enhancing the range of electric vehicles [21]. Bayir et al. utilized regenerative braking methods to recover energy from electric vehicles during downhill descents [22]. Shu et al. introduced a constant-speed downhill regenerative braking control strategy for hybrid electric vehicles. This approach significantly enhances the regenerative braking energy recovery rate, optimizes the battery temperature rise curve, and improves the battery charging rate [23].

In summary, the existing research on inspection robots has some shortcomings, primarily in the following aspects: difficulty in controlling the speed during downhill descents of inspection robots and the lack of effective online energy replenishment methods. To address the aforementioned issues, this paper proposes an energy-efficient speed control method for downhill descents of inspection robots, combining feedback braking and energy consumption braking. By analyzing the forces acting on the inspection robot during downhill descents along power transmission lines, researchers determined the driving

torque of both the front and rear wheels of the robot. To achieve speed control and energy recovery, a scheme was devised that combines front-wheel feedback braking with rear-wheel energy consumption braking. This approach involves adjusting the PWM duty cycle of the downhill speed control system to achieve energy consumption-based speed control for the robot's rear walking wheels [24]. During downhill descents of the inspection robot, energy recovery from the front walking wheels is achieved using feedback braking. Meanwhile, to optimize the speed of the rear wheels, a fuzzy PID control method is employed. The effectiveness of this method in recovering feedback braking energy during steady downhill descents of the inspection robot is validated through simulations and simulated high-voltage line experiments.

## 2. Structure and Operation Route of an Inspection Robot

Currently, the majority of inspection robots studied both domestically and internationally adopt a dual-arm or triple-arm wheel-arm compound structure to perform inspection tasks along predefined routes [24–28]. In this study, a dual-arm wheel-arm compound-structure inspection robot was used as the experimental object. Figure 1 displays the configuration diagram of the high-voltage transmission line inspection robot, which comprises walking wheels, pressing wheels, mechanical arms, mobile joints, and a control box. The two mechanical arms are connected to the control box through mobile joints. The inspection robot is driven by the walking part of the mechanical arms, which can adapt to different line inclinations by adjusting the gripping force between the two arm-pressing wheels and the ground wire. On a gentle slope, a light grip on the ground wire ensures the safety of the robot. Conversely, on a steeper slope, the gripping force between the pressing wheels and the ground wire is increased to prevent the robot's walking wheels from slipping.

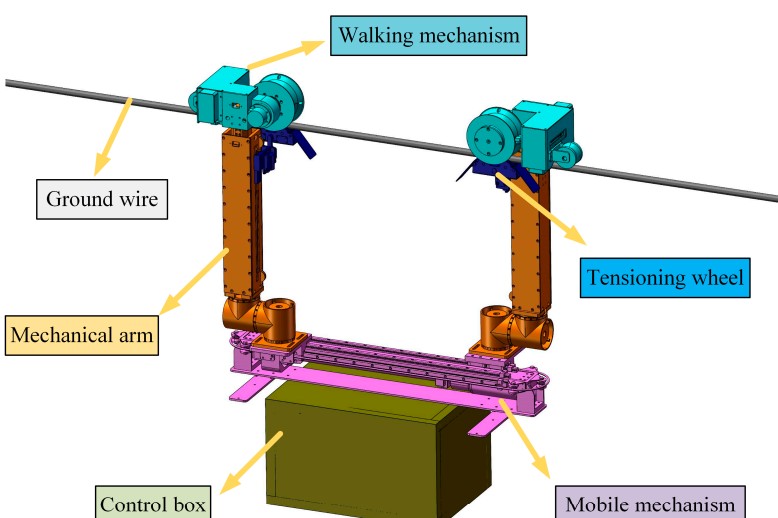

**Figure 1.** Structure diagram of a high-voltage line inspection robot.

As shown in Figure 2, high-voltage transmission towers divide the power lines into multiple spans, and the power lines between adjacent towers form a catenary shape under the influence of gravity. The typical inspection process of the robot along the power line involves descending first and then ascending. The distance and maximum slope angle of the descent are determined by the distance and height difference between the adjacent towers. The tower poles secure the power cables by clamping them with suspension clamps. Typically, for spans greater than 120 m, vibration dampers are added at the suspension points between two towers to protect the safety of the power lines. In inspection areas such as high mountain valleys and pristine forests, the terrain is complex, and the distance and height difference between adjacent towers are usually significant. This provides an opportunity to implement energy recovery for the robot during downhill segments. In

conclusion, the inspection robot is capable of performing inspection tasks in complex terrain environments. Additionally, it can utilize the potential energy from downhill segments to recover energy, thereby enhancing the endurance of the inspection robot.

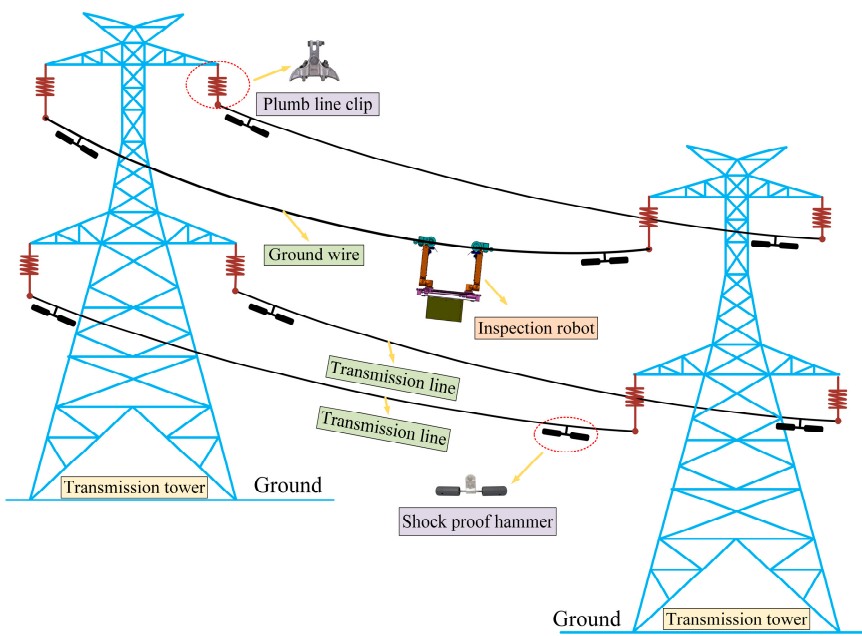

**Figure 2.** Simulation of the operating environment for inspection robots.

## 3. Force Analysis and Speed Control System for Inspection Robots Descending along Overhead Power Lines

### 3.1. Parametric Equations for Overhead Catenary Lines

High-voltage transmission tower construction takes place in complex environments, often found in mountainous and hilly areas. There is a significant span and height difference between the towers, causing the overhead power lines to form a catenary shape under the influence of gravity. This situation is referred to as uneven-height towers. Overhead power lines have relatively low stiffness, making them susceptible to deformation under the influence of gravity. However, they possess high tensile strength and can withstand significant pulling forces. Therefore, it is reasonable to consider overhead power lines as long flexible ropes. This flexible rope can only withstand tension forces and cannot be subjected to bending moments. The load distribution on the line is uniform and in the same direction. There are generally three methods for solving the curve equation of high-voltage power cables: the two-variable function curve method, the parabolic curve method, and the catenary curve method [29]. The catenary curve method can accurately simulate the shape of the cable under the influence of gravity and has a faster computation speed. Therefore, this paper will use the catenary curve method to solve the curve equation of the high-voltage power cable.

Figure 3 illustrates a schematic diagram of an uneven-height suspension point catenary line. The arc *AB* represents the downhill section of the high-voltage power line, which exhibits a catenary shape due to the combined effect of gravity and tension. Points *A* and *B* represent the suspension points of the power line on adjacent high-voltage transmission towers. Point *A* corresponds to the high point of the downhill section, while point *B* marks the low point. The catenary line *AB* has a horizontal distance of *l* and a vertical height difference of *h*. It is subjected to a uniform distributed load of $\gamma$ (N/m·m$^2$) due to the influence of gravity, with the direction vertically downward. Point *O* corresponds to the lowest point of the catenary line, where the stress applied at the two ends is denoted as $\sigma_A$ and $\sigma_B$. To facilitate the analysis, we establish a coordinate system in Figure 3, with the low point *B* on the right side chosen as the origin. The *x*-axis represents the horizontal direction,

and the *y*-axis represents the vertical direction. Taking a segment *MN* for force analysis, Figure 4 illustrates the force analysis diagram of the catenary line segment *MN*. Based on the principle of force equilibrium, we have the following relationships:

$$\sum X = 0, (\sigma_x + d\sigma)\cos(\alpha + \Delta\alpha) = \sigma_x \cos\alpha = \sigma_0 \tag{1}$$

$$\sum Y = 0, (\sigma_x + d\sigma)\sin(\alpha + \Delta\alpha) = \sigma_x \sin\alpha + \gamma dl \tag{2}$$

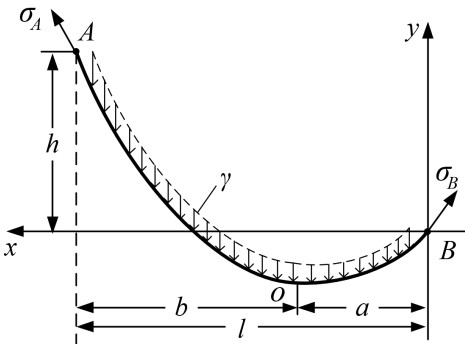

**Figure 3.** Uneven-height suspension point catenary line.

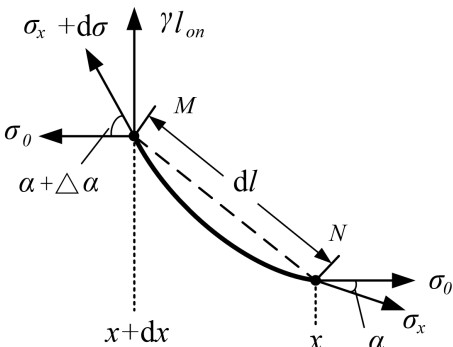

**Figure 4.** Force analysis diagram of catenary line segment MN.

Dividing Equation (1) by Equation (2), according to Lagrange's theorem, we obtain:

$$\tan'\alpha \cdot \frac{d\alpha}{dx} = \frac{\gamma}{\sigma_0}dl \tag{3}$$

The slope of a point on a catenary curve is denoted by $\alpha$, and the slope can be expressed as $\tan\alpha = dy/dx$. Taking the derivative with respect to *x*, we have:

$$\tan'\alpha\frac{d\alpha}{dx} = \frac{d^2y}{dx^2} = y'' \tag{4}$$

$$dl = \sqrt{1 + (\frac{dy}{dx})}dx \tag{5}$$

Substituting Equations (4) and (5) into Equation (3), we obtain:

$$y'' = \frac{\gamma}{\sigma_0}\sqrt{1 + y'}dx \tag{6}$$

Substituting $\tan\alpha = dy/dx$ into Equation (6), we obtain:

$$\sec\alpha d\alpha = \frac{\gamma}{\sigma_0}dx \tag{7}$$

Integrating both sides simultaneously yields:

$$\begin{cases} x = \frac{\sigma_0}{\gamma}(\ln|\sec\alpha + \tan\alpha|) + C_1 \\ y = \frac{\sigma_0}{\gamma}\sec\alpha + C_2 \end{cases} \tag{8}$$

In Equation (8), $\sigma_0$ represents the horizontal stress experienced by the catenary curve, which is related to the material of the overhead cable and the surrounding environment. The specific load $\gamma$ is calculated as $9.8\,P_1/A$, where $P_1$ represents the mass per unit length of the high-voltage transmission cable $(\text{kg}/\text{m}^3)$, and $A$ represents the cross-sectional area of the cable $(\text{mm}^2)$. In this context, $C_1$ and $C_2$ represent integration constants, which acquire specific values based on the "origin" position of the coordinate system. By plugging the coordinates of characteristic points $A$, $B$, and $O$ from the graph into the parameter equation, we can obtain the parameters $C_1$ and $C_2$:

$$\begin{cases} C_1 = \frac{l}{2} - \frac{\sigma_0}{\gamma} arcsh\frac{h}{\frac{2\sigma_0}{\gamma}sh\frac{\gamma l}{2\sigma_0}} \\ C_2 = -\frac{\sigma_0}{\gamma}ch(\frac{\gamma l}{2\sigma_0} - arcsh\frac{h}{\frac{2\sigma_0}{\gamma}sh\frac{\gamma l}{2\sigma_0}}) \end{cases} \tag{9}$$

Substituting Equation (9) into Equation (8), we obtain the parameter equation for the catenary curve of the downhill section of the inspection robot.

$$\begin{cases} x = \frac{\sigma_0}{\gamma}(\ln|\sec\alpha + \tan\alpha| - arcsh\frac{h}{\frac{2\sigma_0}{\gamma}sh\frac{\gamma l}{2\sigma_0}}) + \frac{l}{2} \\ y = \frac{\sigma_0}{\gamma}(\sec\alpha - ch(\frac{\gamma l}{2\sigma_0} - arcsh\frac{h}{\frac{2\sigma_0}{\gamma}sh\frac{\gamma l}{2\sigma_0}})) \end{cases} \tag{10}$$

*3.2. Force Analysis of Inspection Robots along Catenary Curves during Downhill Descent*

During the downhill descent of an inspection robot along an overhead catenary ground wire, the front and rear walking wheels maintain constant contact with the ground wire. Assuming that the front and rear walking wheels do not deform and have the same diameter, the center trajectory of the front and rear walking wheels is an equidistant curve of the ground wire. Figure 5 depicts the motion state and force situation of the inspection robot during downhill movement along the high-voltage transmission line. In the figure, $L$ represents the high-voltage transmission line, $L_K$ represents the trajectory curve of the inspection robot's walking wheels, and curve $L_K$ is an equidistant curve of curve $L$ with a distance equal to the radius $R$ of the walking wheels. During the downhill descent of the inspection robot, the rotation center of the rear walking wheel is $O_1(x_{K1}, y_{K1})$, making an angle $\alpha_1$ with curve $L$. The rotation center of the front walking wheel is $O_2(x_{K2}, y_{K2})$, making an angle $\alpha_2$ with curve $L$. From the diagram, it can be observed that $\alpha_1 > \alpha_2$.

Let $\vec{n}(\alpha)$ be the unit normal vector of curve $L$ that is:

$$\vec{n}(\alpha) = \frac{1}{\sqrt{x'^2 + y'^2}}[-y', x']^T \tag{11}$$

The parametric equation of curve $L_K$ can be obtained based on Equation (10) as:

$$\begin{cases} x_k = \frac{\sigma_0}{\gamma}(\ln|\sec\alpha + \tan\alpha| - arcsh\frac{h}{\frac{2\sigma_0}{\gamma}sh\frac{\gamma l}{2\sigma_0}}) + \frac{l}{2} - R\sin\alpha \\ y_k = \frac{\sigma_0}{\gamma}(\sec\alpha - ch(\frac{\gamma l}{2\sigma_0} - arcsh\frac{h}{\frac{2\sigma_0}{\gamma}sh\frac{\gamma l}{2\sigma_0}})) + R\cos\alpha \end{cases} \tag{12}$$

Considering the inspection robot as a rigid body, during the downhill process without deformation, the distance "$l$" between the front and rear walking wheels has the following relationship with the coordinates $O_1$ and $O_2$:

$$(x_{K1} - x_{K2})^2 + (y_{K1} - y_{K2})^2 = l^2 \tag{13}$$

According to Equation (13), the corresponding coordinates of the other rotational center can be determined for any given coordinate $O_1$ or $O_2$. This means that when the robot is moving downhill along a high-voltage power transmission line if the horizontal coordinates of the center of any wheel are known relative to the starting point, the coordinates of the centers of the front and rear wheels can be obtained. According to geometric relationships, the inspection robot forms an angle $\theta$ with the horizontal ground.

$$\theta = \arctan\frac{y_{K2} - y_{K1}}{x_{K2} - x_{K1}} \tag{14}$$

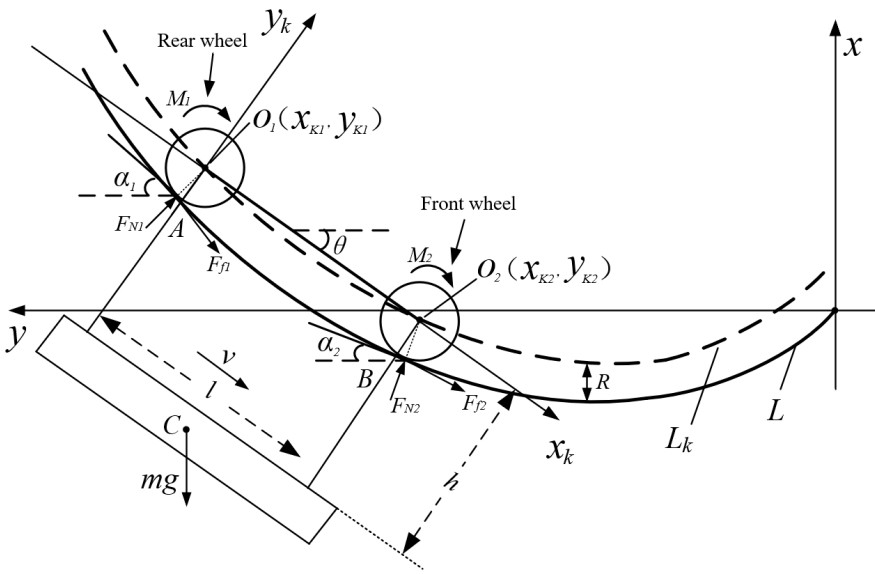

**Figure 5.** Force diagram of the inspection robot during downhill descent along the catenary wire.

Ignoring the wind resistance and internal friction of the walking wheels during the downhill process, the force characteristics of the inspection robot during downhill motion can be analyzed. $F_{N1}$, $M_1$, and $F_{f1}$ represent the normal pressure, driving torque, and rolling friction, respectively, acting on the rear walking wheel. $F_{N2}$, $M_2$, and $F_{f2}$ represent the normal pressure, driving torque, and rolling friction, respectively, acting on the front walking wheel. $\mu$ is the rolling friction coefficient, and point $C$ represents the center of mass. In the analysis of the overall mechanical structure force balance when the inspection robot is moving at a constant speed downhill, it can be obtained:

$$\begin{cases} F_{f1} = \mu F_{N1} \\ F_{f2} = \mu F_{N2} \\ F_{N1}\cos(\alpha_1 - \theta) + F_{f1}\sin(\alpha_1 - \theta) - F_{N2}\cos(\theta - \alpha_2) + F_{f2}\sin(\theta - \alpha_2) - mg\sin\theta = 0 \\ F_{N1}\sin(\alpha_1 - \theta) - F_{f1}\cos(\alpha_1 - \theta) + F_{N2}\sin(\theta - \alpha_2) + F_{f2}\cos(\theta - \alpha_2) - mg\cos\theta = 0 \\ M_1 - F_{f1}R\cos(\alpha_1 - \theta) - F_{f2}R\cos(\theta - \alpha_2) - F_{N2}l\cos(\theta - \alpha_2) - mgh\sin\theta + \frac{1}{2}mgl\cos\theta = 0 \\ M_2 - F_{f1}R\cos(\alpha_1 - \theta) - F_{f2}R\cos(\theta - \alpha_2) - F_{N1}l\cos(\alpha_1 - \theta) - mgh\sin\theta - \frac{1}{2}mgl\cos\theta = 0 \end{cases} \tag{15}$$

In Equation (15), $\alpha_1$ and $\alpha_2$ represent the angles of inclination between the rear walking wheel and the front walking wheel, respectively, and the contact point with the cable during the downhill process of the inspection robot. $\theta$ represents the overall tilt angle of the inspection robot. When the horizontal distance between the rear wheel's walking center and the starting point is determined, the positions of the front and rear wheels and the coordinates of the contact point between the walking wheels and the cable can be determined using Equations (12) and (13). Using Equation (14), the overall tilt angle of the inspection robot can be obtained. Combining this with Equation (15), the variation trend of

the driving torque of the front and rear wheels of the robot with respect to the horizontal walking distance x can be determined.

### 3.3. Analysis of the Driving Torque of an Inspection Robot's Front and Rear Wheels during Downhill Travel

During the study of the torque variations of the front and rear wheels of an inspection robot, while traveling downhill along a power transmission line, the chosen cable model for the 220 kV high-voltage power transmission line is LGJ-GB1179-83. The cable has an outer diameter of 13.6 mm, a horizontal stress $\sigma_0$ of 100 N/mm$^2$, a specific weight $\gamma$ of $26.34 \times 10^{-3}$ N/mm$^2$, a distance between high-voltage transmission towers $L_1$ of 200 m, a height difference between high-voltage transmission towers $H$ of 30 m, the mass of the inspection robot is 40 kg, and the diameter of the inspection robot's walking wheels is $d = 75$ mm. By selecting the horizontal distance $x$ (0 m to 200 m) from the rear wheel to the starting point as the independent variable, we can determine the catenary shape in this example, as shown in Figure 6. The trend of the angle $\theta$ between the robot's body and the horizontal direction as a function of distance $x$ is depicted in Figure 7. The relationship between the driving torque of the front and rear wheels of the robot during downhill travel and the horizontal walking distance $x$ is illustrated in Figure 8.

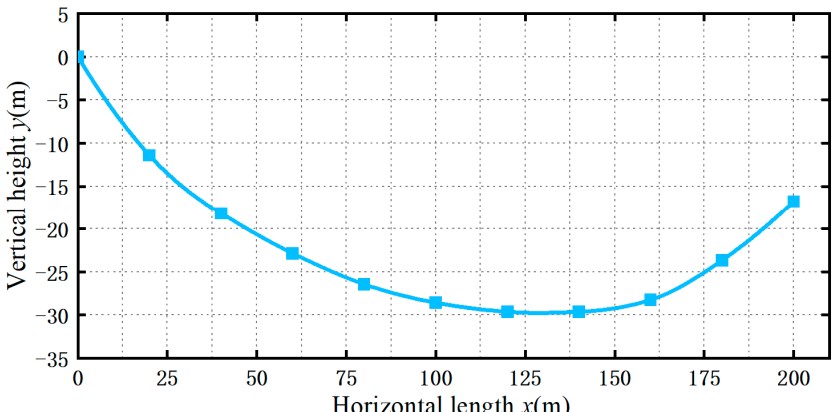

**Figure 6.** Catenary curve of flexible cable.

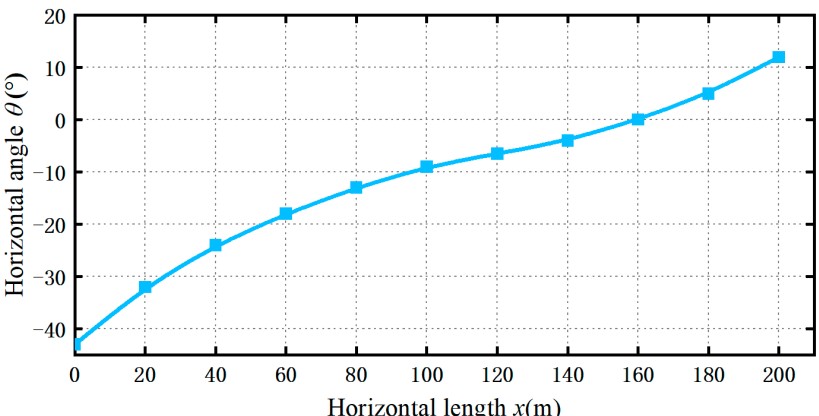

**Figure 7.** Variation trend of the angle $\theta$ between the robot's body and the horizontal direction with distance $x$.

According to Figure 7, it can be observed that the angle between the robot's body and the horizontal direction gradually increases from negative to positive as it walks along the cable. At $x = 171$ m, the cable angle becomes 0, indicating that the cable is horizontal at that point. The cable has its maximum inclination at a higher suspension point, with a maximum angle of 43 degrees. According to Figure 8, it can be observed that during the

downhill process of the inspection robot, the driving torque of the rear wheel is greater than that of the front wheel. However, the variation in the driving torque of the rear wheel is smaller compared to the variation in the driving torque of the front wheel. The direct current (DC) motor energy consumption braking mode for the walking wheels can be controlled at any speed, while the regenerative braking mode requires operating within a certain speed range to achieve an effective braking effect [30]. In order to ensure the safety of the inspection robot during the downhill process, the speed of the entire downhill process must be controllable. In conclusion, during the downhill of the inspection robot, the rear walking wheel of the robot adopts energy consumption braking for speed control, while the front walking wheel utilizes regenerative braking to recover energy during the downhill descent.

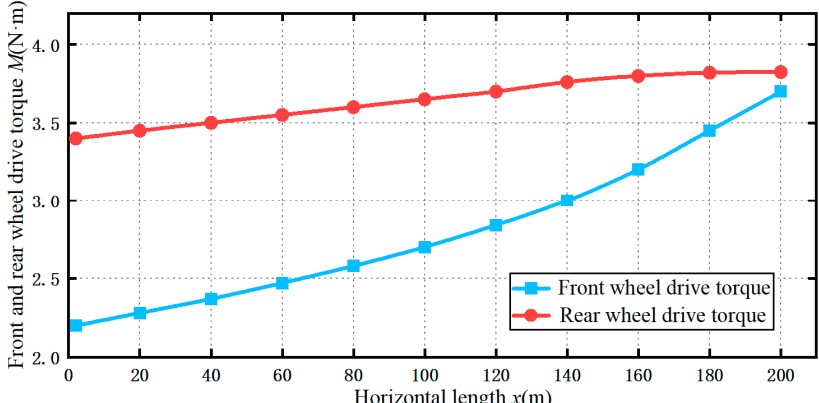

**Figure 8.** Relationship between the driving torque of the front and rear wheels of the robot and the horizontal walking distance *x* during downhill travel.

### 3.4. An Inspection Robot's Walking Wheel Motor Downhill Speed Control System

Figure 9 depicts the structure diagram of the robot's energy consumption control and regenerative energy feedback system. This system consists of the driving section, the rear wheel energy consumption control section, and the front wheel regenerative braking section. The driving section consists of a supercapacitor, a lithium battery, a driver, and a switching circuit. The lithium battery provides energy to the driver and switching circuit, which then controls the motors of the front and rear wheels of the inspection robot. The supercapacitor transfers the recovered energy to the lithium battery. The rear wheel energy consumption control section consists of a rear wheel DC motor, a pulse width modulator, and an energy consumption resistor. The energy consumption resistor can dissipate the energy during the downhill process of the rear walking wheel, enabling the control of the inspection robot's downhill speed. The front wheel regenerative braking section consists of a front wheel DC motor, a reverse diode, and a current sensor. During the downhill process of the inspection robot, when the current sensor detects a reverse current in the circuit of the front walking wheel, the regenerative braking energy generated by the front wheel DC motor during the descent is recovered using the supercapacitor. The reverse diode allows energy to flow only in one direction, from the front wheel DC motor to the supercapacitor. Figure 10 shows the schematic diagram of the energy consumption control circuit for the inspection robot's rear wheel DC motor. In the diagram, $E_1$ represents the electromotive force generated during the downhill process of the rear walking wheel. $R_a$ denotes the energy consumption resistor, while $R_z$ and $L_a$ represent the armature winding's internal resistance and armature inductance of the rear walking wheel's DC motor, respectively. The controller adjusts the duty cycle of the PWM (pulse width modulation) wave to control the opening and closing degree of the insulated gate bipolar transistor *M*, thereby regulating the speed of the inspection robot's rear wheel motor. Figure 11 shows the schematic diagram of the regenerative braking circuit for the inspection robot's front wheel. In the diagram, $i_a$ represents the current in the circuit of the front wheel walking motor. $D_1$ and $D_2$ are reverse

diodes. When the inspection robot operates on overhead power lines, there are frequent uphill and downhill movements. To avoid damage to the lithium battery caused by the large current charging and discharging of the front wheel motor during downhill movement, a composite energy storage model is proposed, which combines a supercapacitor in parallel with the lithium battery. This composite energy storage model adopts a hybrid diode structure, where the supercapacitor and the lithium battery are directly connected across the terminals of the front wheel motor via reverse diodes, ensuring a unidirectional flow of energy through the one-way conduction property of the reverse diodes. The lithium battery serves as the primary power source to supply power to the walking wheel motor, while the supercapacitor recovers the regenerative braking energy generated by the front walking wheel motor during downhill movement. The recovered energy is transferred to the lithium battery through a bidirectional DC/DC converter for storage and management.

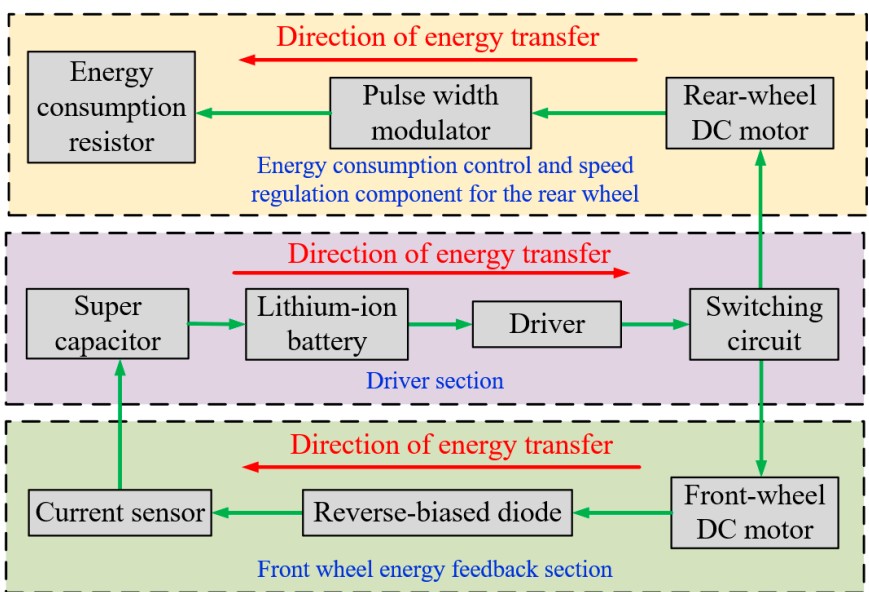

**Figure 9.** Structure of the robot's energy consumption control and regenerative energy storage system.

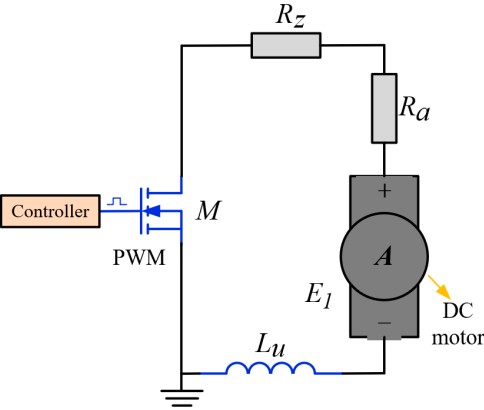

**Figure 10.** Schematic diagram of the energy consumption control circuit for the rear wheel.

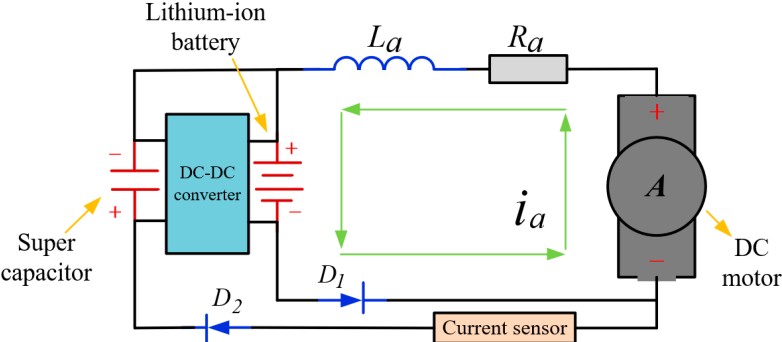

**Figure 11.** Schematic diagram of the regenerative energy storage circuit for the front wheel.

The SOC (state of charge) is one of the key parameters used to describe the remaining capacity of a battery. In order to quantify the amount of energy recovered by the supercapacitor during the regenerative braking of the inspection robot during downhill movement, the SOC of the supercapacitor is defined as the ratio of the energy at a supercapacitor terminal voltage of $V_{cap}$ to the corresponding energy at the fully charged rated voltage $V_e$ of the supercapacitor. The SOC of the supercapacitor is given by:

$$\text{SOC} = \frac{1}{2}CV_{cap}^2 / \frac{1}{2}CV_e^2 = V_{cap}^2 / V_e^2 \tag{16}$$

From Equation (16), it can be observed that the SOC of the supercapacitor has a quadratic relationship with its terminal voltage $V_{cap}$. When $V_{cap}/V_e < 0.5$, the energy discharge efficiency of the supercapacitor is relatively low. Therefore, the cutoff voltage of the supercapacitor is set to half of its rated voltage. In conclusion, during the downhill process of the inspection robot, the current sensing module is used to measure the magnitude and direction of the walking wheel motor current. When the current direction in the circuit is opposite to the non-braking state current direction of the inspection robot, the relay is activated, allowing the supercapacitor to recover regenerative braking energy. The SOC monitoring module continuously monitors the SOC value of the supercapacitor in real time. With a rated voltage of 24 V for the supercapacitor, when the SOC value of the supercapacitor exceeds 95%, the relay is disengaged, interrupting the energy-recovery circuit.

## 4. Control System for an Inspection Robot along an Overhead Ground Line Downhill

### 4.1. Fuzzy PID Controller Structure

Through the analysis of the forces acting on the inspection robot as it descends along the transmission ground wire, it is evident that the driving torque of the front and rear walking wheel motors varies with changes in the line slope during the downhill process. The downhill speed control system for the robot is a non-linear time-varying system, making speed control challenging. Additionally, the speed of the robot's walking wheels during downhill descents is also influenced by external environmental factors such as cable material, surface corrosion level, rain, snow, and wind force. It is not feasible to control the speed of the walking wheel motors through precise mathematical models. Therefore, it is necessary to continuously adjust the driving torque of the front and rear walking wheel motors of the robot based on the real-time slope of the cable and the real-time speed of the walking wheels during the robot's downhill movement.

The PID control adjusts the output of the control variable based on the current error, the accumulated error, and the rate of change of the error to enhance system stability. Traditional PID control lacks parameter-adaptive tuning capabilities; its parameter tuning must be relative to a known system, and the tuned parameters may only be applicable to a specific operating condition of the inspection robot. As a result, fixed PID parameters inherently cannot meet the requirements of non-linear and time-varying control systems. Fuzzy control is a control method based on fuzzy logic and is used to deal with systems that

are difficult to precisely model and control. Fuzzy control uses fuzzy sets and fuzzy logic to handle uncertainty and vagueness within a system, enabling the controller to flexibly adapt to the complex changes within the system.

Fuzzy PID control combines the advantages of fuzzy control and PID control, integrating fuzzy set theory and fuzzy inference techniques from fuzzy logic into traditional PID control to enhance the robustness of control systems against non-linear, time-varying, and uncertain factors. The downhill speed control system of the inspection robot is susceptible to uncertain factors such as cable corrosion and wind force. To enhance the stability and control accuracy of the inspection robot's downhill speed, an adaptive fuzzy PID control method will be employed to control the downhill speed of the robot's walking wheel motor. This aims to achieve uniform downhill speed for the inspection robot along flexible cables. Figure 12 illustrates the principle of fuzzy PID control for the downhill speed of the robot's walking wheels.

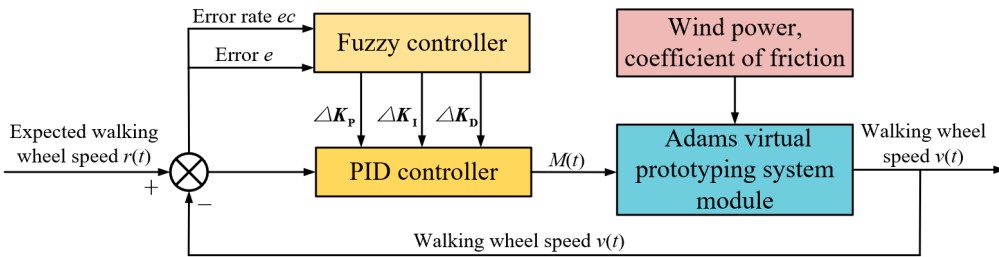

**Figure 12.** Schematic diagram of the fuzzy PID control principle for the downhill speed of a robot's walking wheels.

As shown in Figure 12, the robot's walking wheel speed v and its rate of change will be used as inputs to the adaptive fuzzy PID controller, while the driving torque M of the robot's walking wheel motor will be the output of the adaptive fuzzy PID controller. In this paper, the downhill motion of the inspection robot is simulated using the Adams virtual prototype. The input value of the simulation is the driving torque $M$ of the walking wheel motor, and the output value is the speed of the walking wheel motor. The output value is used as negative feedback to the adaptive fuzzy PID controller, dynamically adjusting the downhill speed of the inspection robot along the overhead ground line. The system is also subjected to external disturbances such as wind force and cable corrosion.

*4.2. Fuzzy PID Controller Design*

The deviation between the desired walking wheel speed, denoted as $r(t)$, and the feedback walking wheel speed, denoted as $v(t)$, is referred to as the error $e$. The rate of change of the error $e$ with respect to relative time is represented as $ec$. These error $e$ and error change rate $ec$ are utilized as the two input variables for the fuzzy controller, with the output variables being the PID controller parameter adjustments $\Delta K_P$, $\Delta K_I$, and $\Delta K_D$. This results in the development of a fuzzy PID controller. In this paper, the fuzzy controller employs seven fuzzy subsets for each of the five variables, namely NB (Negative Big), NM (Negative Medium), NS (Negative Small), ZO (Zero), PS (Positive Small), PM (Positive Medium), and PB (Positive Big). Gaussian-type membership functions are chosen for the linguistic variables of error $e$ and error change rate $ec$. The setpoint for the inspection robot's downhill walking wheel motor speed is set at 6000 r/min, with speed fluctuations not exceeding ±10% once the set speed is reached. Thus, the domain for the error $e$ is set to $(-600, 600)$, and the domain for error change rate $ec$ is set to $(-300, 300)$. For the output variables $\Delta K_P$, $\Delta K_I$, and $\Delta K_D$ of the fuzzy control, triangular membership functions are employed. The domain for output variable $\Delta K_P$ is set to $(-6, 6)$, the domain for output variable $\Delta K_I$ is set to $(-12, 12)$, and the domain for output variable $\Delta K_D$ is set to $(-6, 6)$. Figures 13 and 14, respectively, depict the membership functions for input variables and output variables.

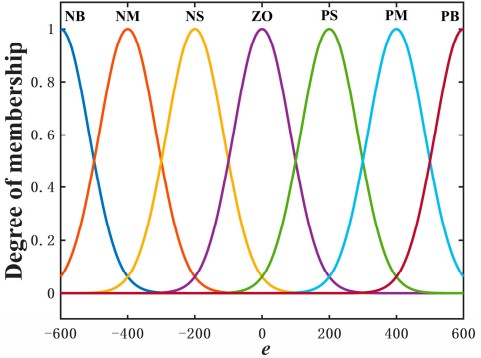 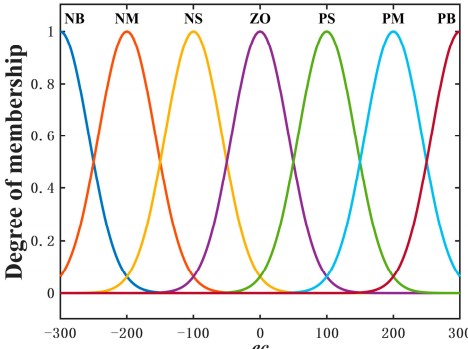

**Figure 13.** The input membership functions for the error *e* and the error change rate *ec*.

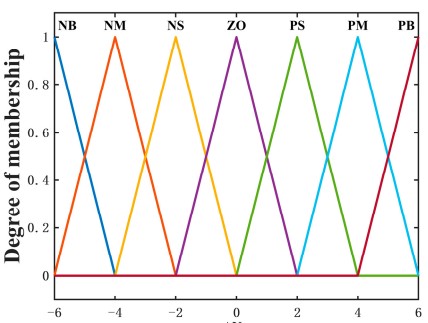 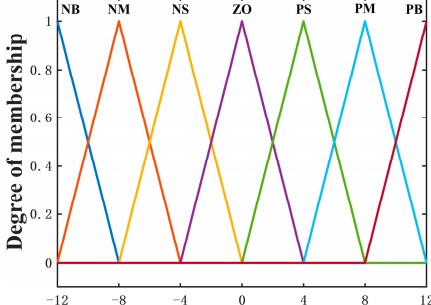 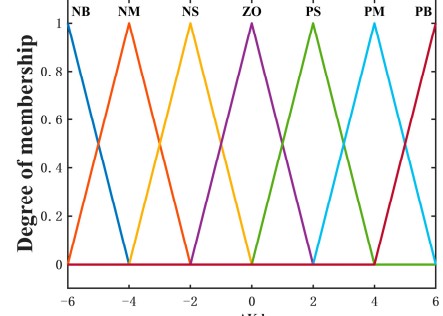

**Figure 14.** The membership functions of the fuzzy control output variables $\Delta K_P$, $\Delta K_I$, and $\Delta K_D$.

Tables 1–3 present the fuzzy rules set based on the downhill characteristics of the inspection robots along flexible cables. These rules were edited in the fuzzy logic designer of MATLAB. By applying these fuzzy rules, output surfaces for $\Delta K_P$, $\Delta K_I$, and $\Delta K_D$ can be obtained, as shown in Figure 15.

**Table 1.** Fuzzy control rule table for $\Delta K_P$.

|  |  | *ec* | | | | | | |
|---|---|---|---|---|---|---|---|---|
|  |  | **NB** | **NM** | **NS** | **ZO** | **PS** | **PM** | **PB** |
|  | NB | PB | PB | PM | PM | PS | ZO | ZO |
|  | NM | PB | PB | PM | PS | PS | ZO | NS |
|  | NS | PM | PM | PM | PS | ZO | NS | NS |
| *e* | ZO | PM | PM | PS | ZO | NS | NM | NM |
|  | PS | PS | PS | ZO | NS | NS | NM | NM |
|  | PM | PS | ZO | NS | NM | NM | NM | NB |
|  | PB | ZO | ZO | NM | NM | NM | NB | NB |

**Table 2.** Fuzzy control rule table for $\Delta K_I$.

|  |  | *ec* | | | | | | |
|---|---|---|---|---|---|---|---|---|
|  |  | **NB** | **NM** | **NS** | **ZO** | **PS** | **PM** | **PB** |
|  | NB | NB | NB | NM | NM | NS | ZO | ZO |
|  | NM | NB | NB | NM | NS | NS | ZO | ZO |
|  | NS | NB | NM | NS | NS | ZO | PS | PS |
| *e* | ZO | NM | NM | NS | ZO | PS | PM | PM |
|  | PS | NM | NS | ZO | ZO | PS | PM | PB |
|  | PM | ZO | ZO | PS | PS | PM | PB | PB |
|  | PB | ZO | ZO | PS | PM | PM | PB | PB |

**Table 3.** Fuzzy control rule table for $\Delta K_D$.

|  |  | ec | | | | | | |
|---|---|---|---|---|---|---|---|---|
|  |  | **NB** | **NM** | **NS** | **ZO** | **PS** | **PM** | **PB** |
| *e* | NB | PB | PB | PM | PM | PS | ZO | ZO |
|  | NM | PB | PB | PM | PS | PS | ZO | NS |
|  | NS | PM | PM | PM | PS | ZO | NS | NS |
|  | ZO | PM | PM | PS | ZO | NS | NM | NM |
|  | PS | PS | PS | ZO | NS | NS | NM | NM |
|  | PM | PS | ZO | NS | NM | NM | NM | NB |
|  | PB | ZO | ZO | NM | NM | NM | NB | NB |

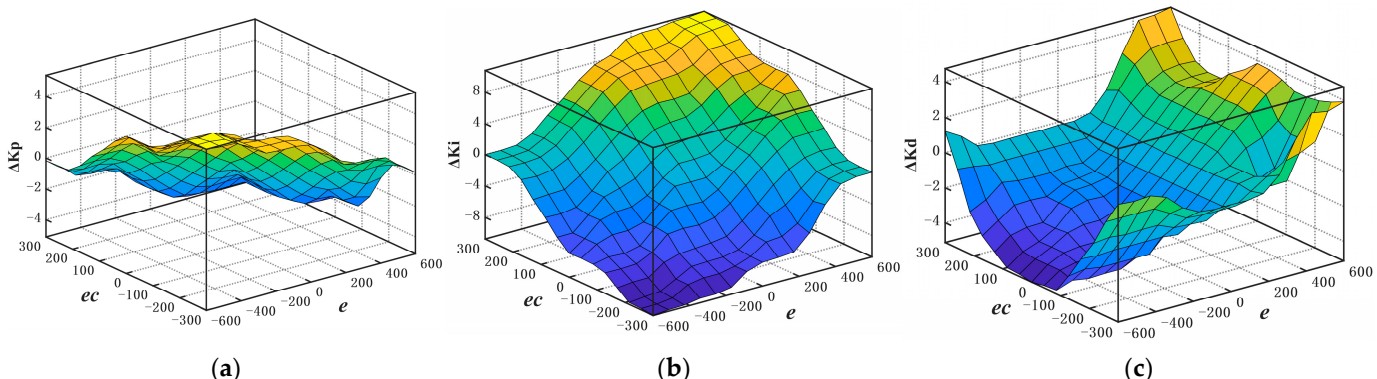

(a)　　　　　　　　　　(b)　　　　　　　　　　(c)

**Figure 15.** Fuzzy PID controller parameter variation surfaces. (**a**) $\Delta K_P$, (**b**) $\Delta K_I$, (**c**) $\Delta K_D$.

The output generated by the fuzzy inference engine is in the form of fuzzy subsets, which need to undergo a defuzzification process to convert them into precise deterministic values. Common methods for defuzzification include the maximum membership value method, centroid method, and weighted average method. Among these, the centroid method is chosen as the defuzzification strategy in this paper due to its advantages of smooth output and avoidance of information loss.

## 5. Experimental Analysis

### 5.1. Building an Adams/Simulink Co-Simulation Platform

The rigid-flex coupling model of the inspection robot along the downhill transmission line, based on the Adams/Simulink simulation platform, consists of two parts: the rigid robot and the flexible cable. The rigid model of the inspection robot is created using SolidWorks and then imported into the Adams project. Different types of joints, such as revolute joints, prismatic joints, and fixed joints, are added based on the functions of different components. For example, revolute joints are added between the walking wheels and the rotating shaft; prismatic joints are established between the pressing wheels and the mechanical arm; and fixed joints are created between the control box and the moving joint, along with the application of driving torques. The flexible cable is converted into a modal neutral file (MNF) using Abaqus 2022 software and then imported into the Adams project. The two ends of the flexible cable are connected by hinges and towers. Constraints are established between the rigid robot model and the flexible cable, and friction forces are set. Figure 16 shows the rigid–flex coupling model of the robot along the downhill flexible cable.

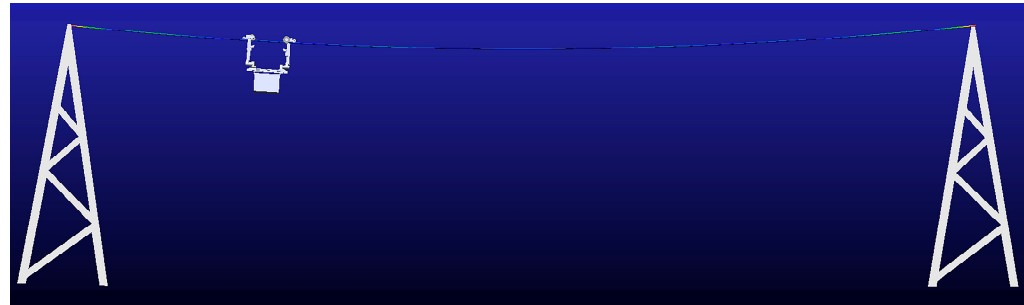

**Figure 16.** Rigid–flex coupling model of the robot along the downhill flexible cable.

As shown in Figure 17, a simulation model for PID control, fuzzy control, and fuzzy PID control was built in the MATLAB/Simulink environment. The fuzzy PID control simulation model was compared with the simulation models of PID control and fuzzy control for comparative simulation experiments. Considering the non-linear and time-varying nature of the inspection robot's downhill speed control system and its susceptibility to external environmental factors like cable corrosion, wear, and wind force, it is challenging to establish an accurate mathematical model to control the walking wheel motor's speed. Hence, a combination of the Ziegler–Nichols tuning method and the ultimate gain method was used to adjust the parameters of the inspection robot's downhill speed control system [31]. The Ziegler–Nichols tuning method adjusts the proportional coefficient $K_P$ first, then the integral coefficient $K_I$, and finally the derivative coefficient $K_D$, to initially determine the range of values for the three parameters of the PID controller. The ultimate gain method involves observing the response curve, considering response speed and overshoot, and gradually increasing the value of the proportional coefficient $K_P$ until the system begins to oscillate. Then, gradually increasing the integral coefficient $K_I$ can reduce the amplitude and frequency of the system's response curve oscillations, enhancing system stability. Finally, adjusting the derivative coefficient $K_D$ further suppresses oscillations and reduces the system's overshoot. Ultimately, the initial values of the three parameters, $K_P$, $K_I$, and $K_D$, for the fuzzy PID controller were set as $K_{P0} = 55$, $K_{I0} = 102$, and $K_{D0} = 50$.

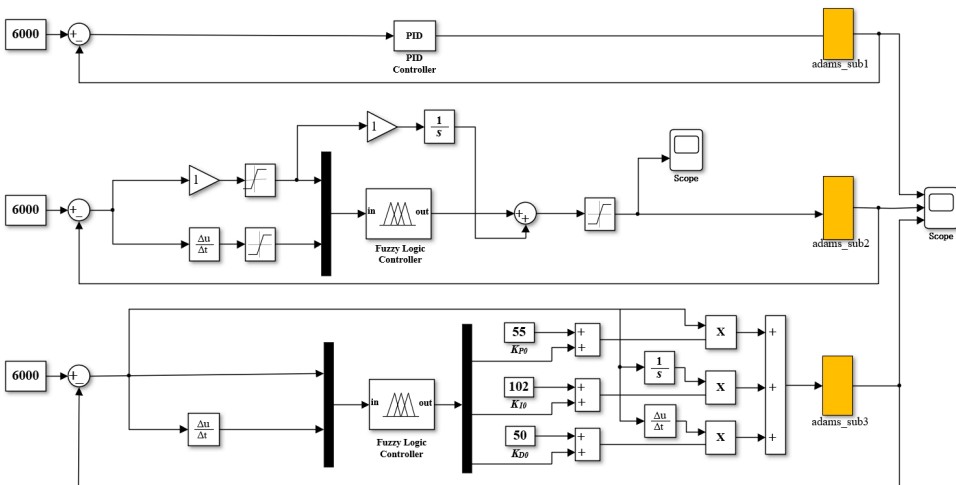

**Figure 17.** Comparative simulation models for PID control, fuzzy control, and fuzzy PID control.

### 5.2. Analysis of Simulation Experimental Results

Due to the complexity of the downhill speed control test for the inspection robot, in order to avoid the actual experiment causing damage to the robot, the feasibility of the control system is verified through joint simulation using Adams/Simulink before the test. The simulation is performed on a rigid–flexible coupled model, with a simulation time of 20 s and 200 simulation steps in Adams. The target wheel speed for the inspection

robot descending along the flexible cable is set to 6000 r/min. The simulation process includes downhill tests along the flexible cable and with inclinations of 10°, 20°, and 30°, respectively. Fuzzy control, PID control, and fuzzy PID control are compared in the experiments. Figure 18 depicts the simulation model of the inspection robot descending along slopes of 10°, 20°, and 30°. Figure 19 illustrates the simulation plot of the robot descending along a flexible cable, while Figure 20 shows the simulation curves of the robot's speed during descents along slopes of 10°, 20°, and 30°.

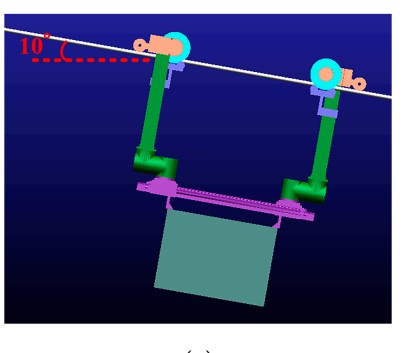

(a)

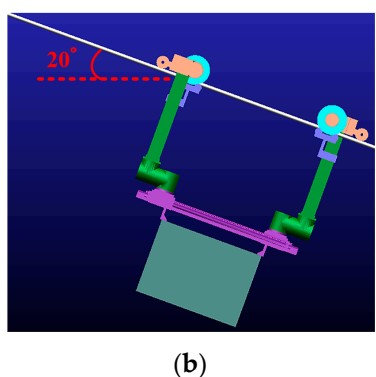

(b)

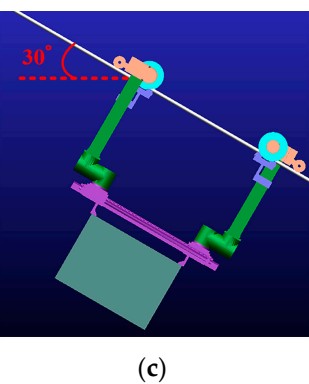

(c)

**Figure 18.** Robot model for downhill slopes of 10°, 20°, and 30°. (**a**) 10°, (**b**) 20°, (**c**) 30°.

From Figure 19, it can be observed that fuzzy control, PID control, and fuzzy PID control all successfully regulate the rear wheel speed of the inspection robot to the desired speed of 6000 r/min during the descent along the flexible cable. In comparison to PID control, fuzzy PID control reduces the overshoot of the rear wheel motor speed by 212 r/min. Furthermore, compared to fuzzy control, fuzzy PID control achieves the desired speed in a shorter transition time, reducing it by 1.8 s. Therefore, overall, fuzzy PID control outperforms fuzzy control and PID control in terms of reducing overshoot and shortening the transition time. In Figure 20, fuzzy PID control is utilized to conduct downhill tests with inclinations of 10°, 20°, and 30° for the inspection robot. The time required for the rear wheel to reach the target speed is 12.28 s, 8.36 s, and 5.62 s, respectively. During the actual operation of the inspection robot, it is typically subjected to external factors such as friction on the cable and wind resistance, which can impact the stability of the system. Therefore, the simulation process needs to consider the influence of external disturbances on the system's stability. The simulation diagram in Figure 21 shows a robot descending along cables with different levels of roughness. In Adams, the friction coefficient $\mu$ between the cables and the walking wheels is set. At Time = 6 s, $\mu$ changes from 0.2 to 0.1, reducing the friction force between the walking wheels and the cables. Consequently, the rotational speed of the walking wheels increases to 6248 r/min. Subsequently, under the influence of fuzzy PID control, the speed gradually decreases and stabilizes to the desired speed. At Time = 14 s, $\mu$ returns to 0.2 from 0.1, increasing the friction force between the walking wheels and the cables. As a result, the rotational speed of the walking wheels decreases to 5736 r/min, and after 2 s, it gradually returns to the desired speed. The simulation curve in Figure 22 depicts the robot's downhill motion under the influence of wind drag. Turbulent wind conditions are introduced from 4 to 16 s. As a result, the rotational speed of the walking wheels fluctuates within the range of 5800 to 6200 r/min due to the wind drag. After 16 s, the rotational speed gradually returns to the desired speed. In summary, the downhill speed control system of the inspection robot can effectively overcome the speed fluctuations caused by cable wear and wind force, thereby ensuring precise control of uniform downhill speed.

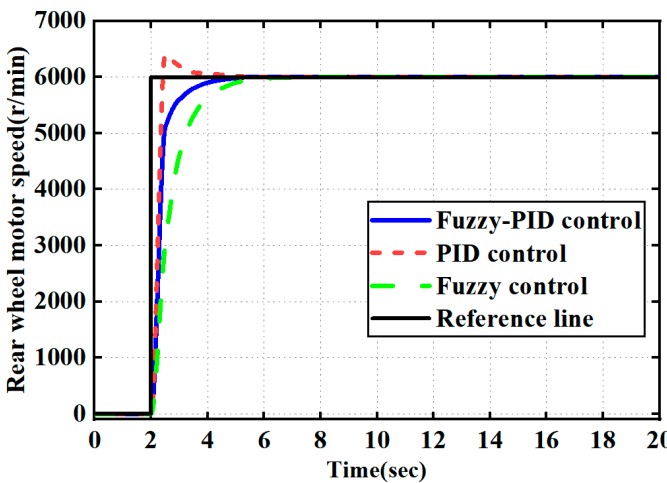

**Figure 19.** Simulation curves of the robot's speed while descending along a flexible cable.

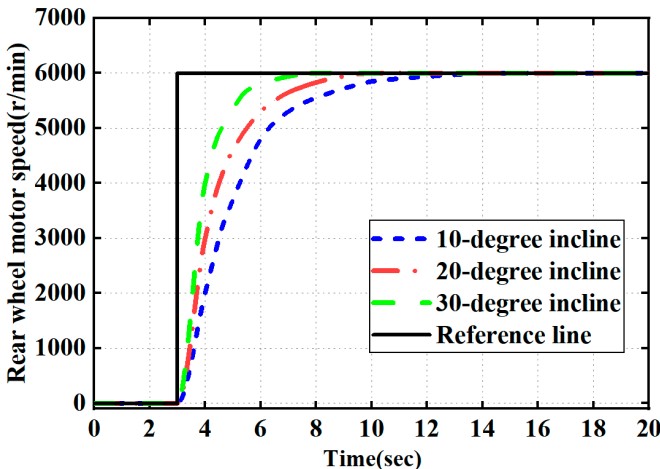

**Figure 20.** Simulation curves of speed while descending along slopes of 10°, 20°, and 30°.

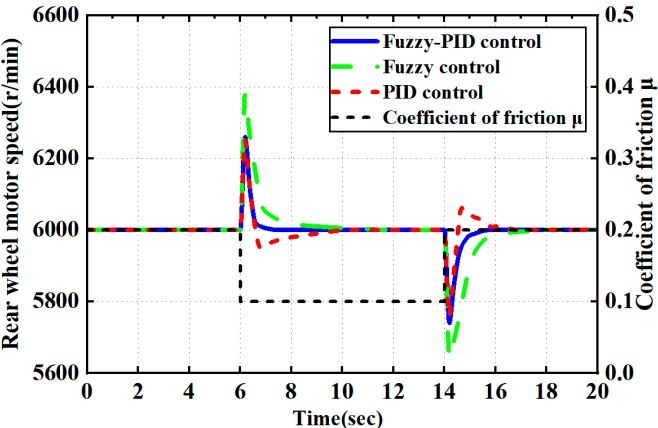

**Figure 21.** Simulation curves of the robot's speed while descending along cables with different levels of roughness.

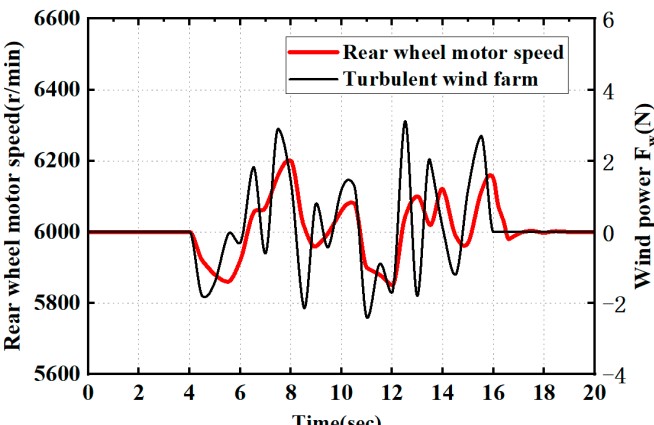

**Figure 22.** Simulation curves of the robot's speed during downhill movement under the influence of turbulent wind conditions.

*5.3. Analysis of Experimental Results*

The designed dual-arm wheeled inspection robot is shown in Figure 23. To simulate the operational environment of the inspection robot, a high-voltage transmission simulation line was constructed indoors. Figure 24 depicts the experimental setup of the inspection robot conducting downhill tests along the transmission cables.

In the downhill testing of the inspection robot along the transmission cables, to verify the effects of cable corrosion and cable wear on the rear wheel speed control, two scenarios were created manually: a section with cable corrosion and a section with cable wear. Through experimentation, the friction coefficients were measured to be 0.312 for the section with cable corrosion and 0.124 for the section with cable wear. Figure 25 shows the speed curve of the robot descending along cables with different levels of roughness. From the graph, it can be observed that when the robot passes through the section with cable corrosion, the increased friction coefficient of the cable leads to a decrease in the rotational speed of the rear walking wheels accompanied by fluctuations. However, the speed gradually returns to the set value after passing through the corroded section. Similarly, when the robot traverses the section with cable wear, the decreased friction coefficient of the cable results in an increase in the rotational speed of the rear walking wheels. Again, the speed gradually returns to the set value after passing through the worn section. To verify the impact of wind drag and body oscillations on the rear wheel speed control during the operation of the inspection robot, an additional wind force was applied to induce body oscillations during the downhill motion. Figure 26 presents the speed curve of the robot during oscillation caused by wind force. From the graph, it can be seen that the body undergoes oscillations within a range of $\pm 10°$ due to the wind force. The rotational speed of the rear walking wheels fluctuates near the target speed while the body is oscillating, indicating controllable speed throughout the process.

Figure 27 depicts the simulation curves of the robot's downhill speed and body sway angle under the influence of crosswind. A wind force device is placed on the side of the downhill path of the inspection robot. This device introduces lateral wind interference during the robot's downhill movement. The changes in the body sway angle are captured through attitude measurement sensors. The crosswind from the wind force device is fan-shaped, with maximum wind force when facing the device and lesser wind force on the sides. The body sway angle of the inspection robot changes as it is affected by the crosswind during the downhill process. The angle initially increases and then decreases. When the inspection robot faces the wind force device directly, the maximum body sway angle reaches 3.17°. Despite the fluctuation in downhill speed caused by crosswind effects, the control system maintains stability, preventing any loss of speed control.

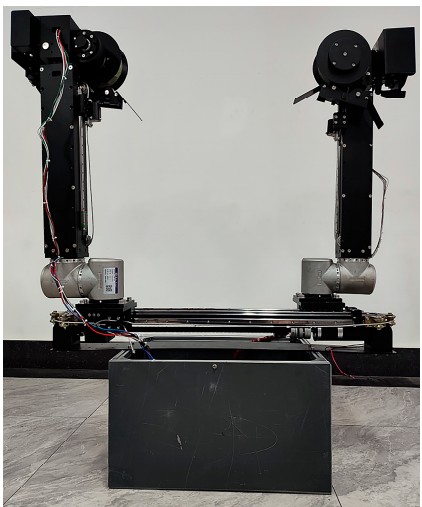

**Figure 23.** Front view of the inspection robot.

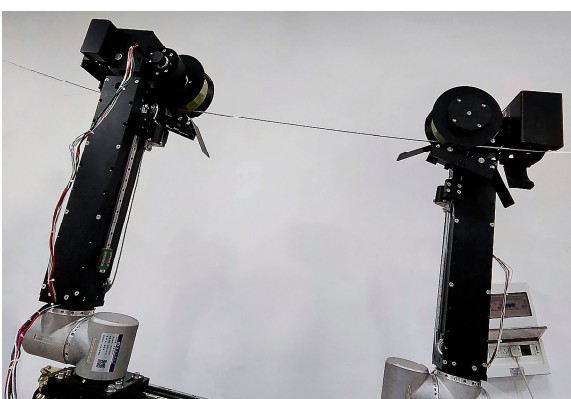

**Figure 24.** Experimental setup of the inspection robot conducting downhill tests along the transmission cables.

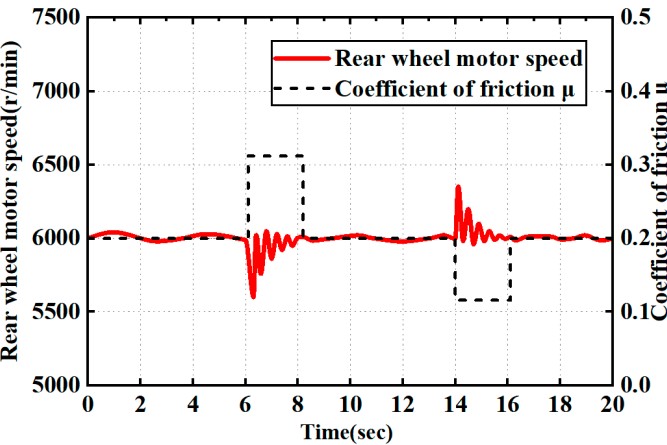

**Figure 25.** Speed curve of the robot descending along cables with different levels of roughness.

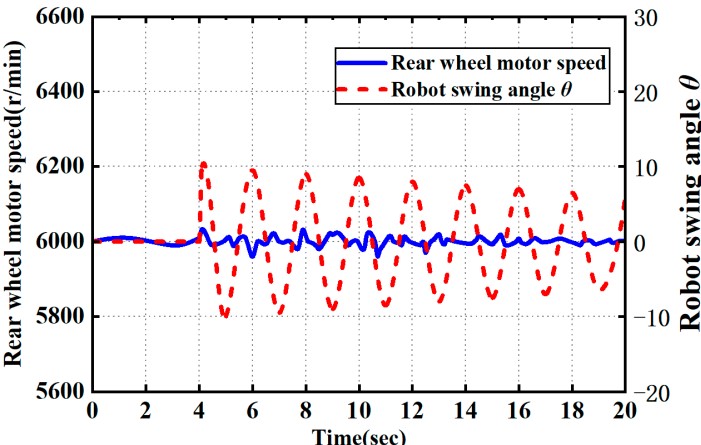

**Figure 26.** Speed curve of the robot during downhill motion while in a swinging state.

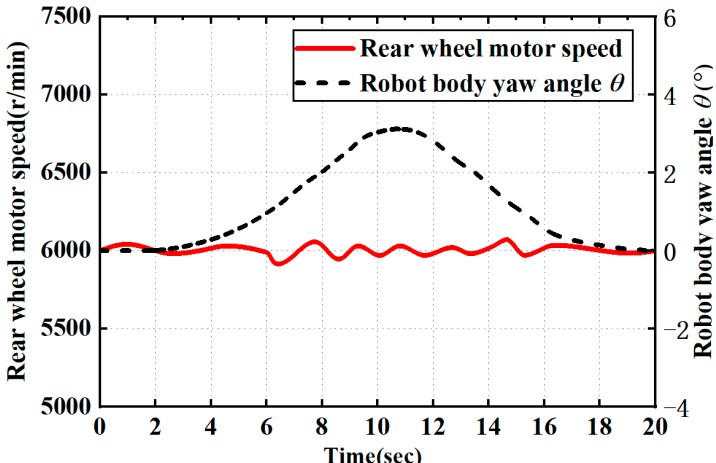

**Figure 27.** Simulation curves of the robot's downhill speed and body sway angle under the influence of crosswind.

To verify the effectiveness of the energy-saving downhill speed control method for the inspection robot's walking wheel motor that combines feedback braking with fuzzy PID control in terms of energy recovery, PID control, fuzzy control, and fuzzy PID control were employed to control the robot's downhill speed along a fixed 30° slope cable. A composite power source was used to recover energy during the downhill descent. Figure 28 illustrates the voltage variation curves of the supercapacitor terminal under PID control, fuzzy control, and fuzzy PID control while descending along the fixed 30° slope cable. Based on the data provided in Table 4, it can be observed that when using fuzzy PID control, the initial terminal voltage of the supercapacitor was 10 V, and after the operation, the terminal voltage was 17.93 V. Using the state of charge (SOC) method, it can be calculated that the supercapacitor SOC increased by 38.45%. With PID control, the initial terminal voltage of the supercapacitor was 10 V, and after the operation, the terminal voltage was 17.06 V, resulting in a SOC increase of 33.17%. Under fuzzy control, the initial terminal voltage of the supercapacitor was 10 V, and after the operation, the terminal voltage was 15.47 V, leading to a SOC increase of 24.19%. In conclusion, when utilizing fuzzy PID control, the increase in supercapacitor SOC was 5.28% higher compared to using PID control and 14.26% higher compared to using fuzzy control. Thus, during the inspection robot's downhill movement along a fixed 30° slope cable, the energy-recovery efficiency achieved through fuzzy PID control surpasses that of PID control and fuzzy control.

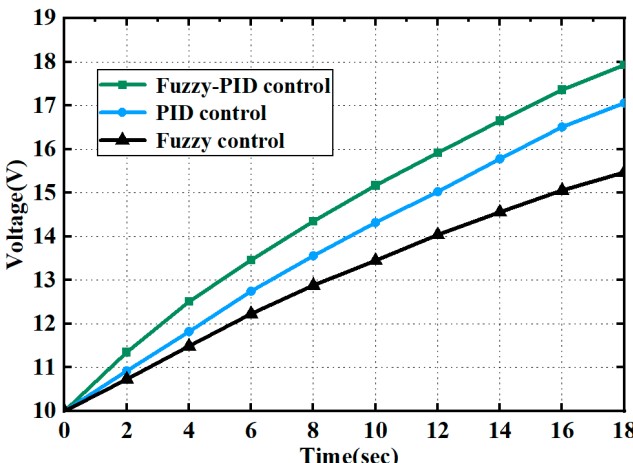

**Figure 28.** Voltage variation curves of the supercapacitor terminal under PID control, fuzzy control, and fuzzy PID control while descending along a fixed 30° slope cable.

**Table 4.** Energy recovery of the supercapacitor under PID control, fuzzy control, and fuzzy PID control while descending along a fixed 30° slope cable.

| Time/s | Voltage of the Supercapacitor under PID Control | Voltage of the Supercapacitor under Fuzzy Control | Voltage of the Supercapacitor under Fuzzy PID Control |
|---|---|---|---|
| 0 | 10.00 | 10.00 | 10.00 |
| 2 | 10.56 | 10.84 | 11.35 |
| 4 | 11.13 | 11.59 | 12.51 |
| 6 | 11.72 | 12.33 | 13.46 |
| 8 | 12.25 | 13.06 | 14.35 |
| 10 | 12.67 | 13.64 | 15.17 |
| 12 | 13.04 | 14.26 | 15.92 |
| 14 | 13.45 | 14.81 | 16.65 |
| 16 | 13.82 | 15.36 | 17.36 |
| 18 | 14.18 | 15.85 | 17.93 |

To investigate the effects of different fixed and variable slopes on the downhill speed control and energy-recovery efficiency of the inspection robot, the fuzzy PID control method was utilized to make the inspection robot descend uniformly along cables with fixed slopes of 10°, 20°, and 30°, as well as a variable slope ranging from 30° to 0°, while recovering energy. Figure 29 illustrates the experimental setup of the robot descending along cables with fixed slopes of 10°, 20°, and 30°. As depicted in Figure 30, the inspection robot was able to achieve the target speed while descending along cables with fixed slopes of 10°, 20°, and 30°, as well as the variable slope of 30° to 0°. The time required to reach the target speed decreased as the slope increased. Figure 31 displays the voltage variation curves of the supercapacitor terminal while the inspection robot descended along cables with fixed slopes of 10°, 20°, and 30°, as well as the variable slope of 30° to 0°. According to the data provided in Table 5, it can be observed that when descending along a fixed 10° slope cable, the supercapacitor's initial terminal voltage was 10 V, and after the operation, the terminal voltage was 17.93 V, resulting in a SOC increase of 38.45%. Similarly, for the fixed 20° slope cable descent, the supercapacitor's initial terminal voltage was 10 V, and the terminal voltage after the operation was 15.85 V, leading to a SOC increase of 26.25%. For the fixed 30° slope cable descent, the supercapacitor's initial terminal voltage was 10 V, and the terminal voltage after the operation was 14.18 V, resulting in a SOC increase of 17.55%. Finally, when descending along the variable slope of 30° to 0°, the supercapacitor's initial terminal voltage was 10 V, and the terminal voltage after the operation was 16.74 V, resulting in a SOC increase of 31.29%. In summary, the efficiency of energy recovery

during downhill movement by the inspection robot increases with the slope of the cable. Moreover, energy recovery efficiency is higher when descending along a fixed 30° slope cable compared to a variable slope cable ranging from 30° to 0°.

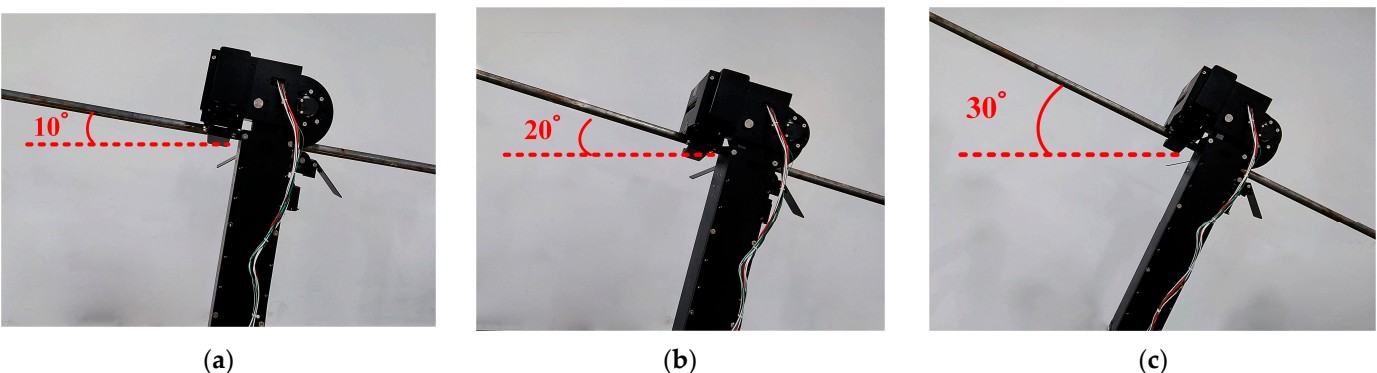

(a)  (b)  (c)

**Figure 29.** Experimental setup of the robot descending along cables with fixed angles of 10°, 20°, and 30°. (**a**) 10°, (**b**) 20°, (**c**) 30°.

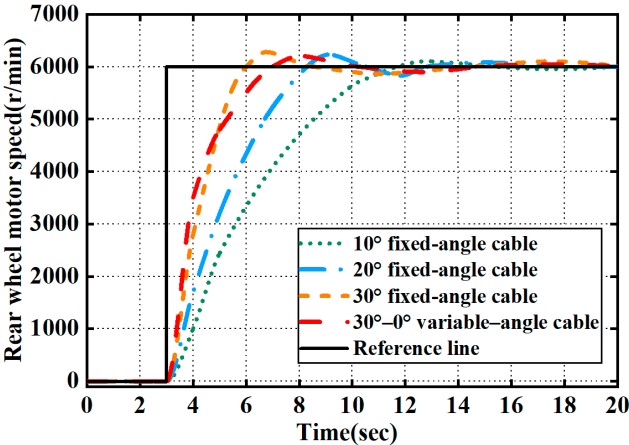

**Figure 30.** Speed curves of the robot while descending along cables with fixed angles of 10°, 20°, and 30°, as well as a variable angle cable ranging from 30° to 0°.

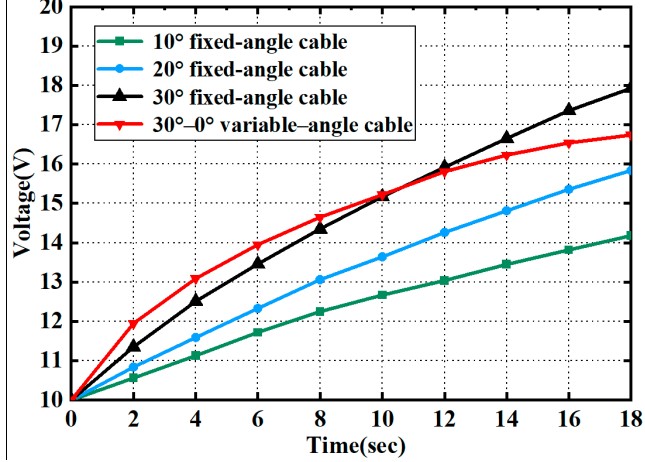

**Figure 31.** Voltage variation curves of the supercapacitor terminal while descending along cables with fixed angles of 10°, 20°, and 30°, as well as a variable angle cable ranging from 30° to 0°.

**Table 5.** Experimental data of supercapacitor energy recovery while descending along cables with fixed angles of 10°, 20°, and 30°, as well as a variable angle cable ranging from 30° to 0°.

| Time/s | Voltage of the Supercapacitor Terminal under Descent along a Fixed 10° Angle Cable | Voltage of the Supercapacitor Terminal under Descent along a Fixed 20° Angle Cable | Voltage of the Supercapacitor Terminal under Descent along a Fixed 30° Angle Cable | Voltage of the Supercapacitor Terminal under Descent along a Variable Angle Cable Ranging from 30° to 0°. |
|---|---|---|---|---|
| 0 | 10.00 | 10.00 | 10.00 | 10.00 |
| 2 | 10.56 | 10.84 | 11.35 | 11.95 |
| 4 | 11.13 | 11.59 | 12.51 | 13.09 |
| 6 | 11.72 | 12.33 | 13.46 | 13.95 |
| 8 | 12.25 | 13.06 | 14.35 | 14.65 |
| 10 | 12.67 | 13.64 | 15.17 | 15.22 |
| 12 | 13.04 | 14.26 | 15.92 | 15.81 |
| 14 | 13.45 | 14.81 | 16.65 | 16.23 |
| 16 | 13.82 | 15.36 | 17.36 | 16.54 |
| 18 | 14.18 | 15.85 | 17.93 | 16.74 |

## 6. Conclusions

(1) By analyzing the forces acting on the robot during downhill motion along a suspended cable, the relationship between the driving torque of the front and rear wheels and the horizontal displacement was determined. Based on this analysis, a speed control and energy-recovery scheme combining front-wheel feedback braking and rear-wheel regenerative braking was designed;

(2) A coupled model incorporating both a rigid robot and a flexible cable was established. The fuzzy PID control algorithm was applied in joint simulations to control the speed of the rear walking wheel during the robot's downhill motion. The results showed that fuzzy PID control outperformed fuzzy control and PID control in effectively mitigating disturbances caused by cable friction and wind drag;

(3) By employing a hybrid power source composed of lithium batteries and supercapacitors, the regenerative braking energy from the front wheels is effectively recovered. When descending along cables with fixed angles of 10°, 20°, and 30°, and a variable angle ranging from 30° to 0°, the supercapacitor's state of charge (SOC) increased by 17.55%, 26.25%, 38.45%, and 31.29%, respectively, demonstrating the efficient absorption of regenerative braking energy during the robot's downhill movement. Moreover, when descending along a fixed 30° angle cable, the fuzzy PID control yielded an increase in supercapacitor SOC of 5.28% compared to PID control and an increase of 14.26% compared to fuzzy control. This confirms that fuzzy PID control outperforms both PID control and fuzzy control in terms of energy recovery efficiency.

**Author Contributions:** Conceptualization, Z.Y. and X.L.; methodology, X.L.; software, C.N.; validation, L.L., W.T. and H.W.; formal analysis, D.Z. (Daode Zhang); investigation, H.L.; resources, D.Z. (Dehua Zhou); data curation, J.K.; writing—original draft preparation, X.L.; writing—review and editing, X.L.; visualization, X.L.; supervision, Z.Y.; project administration, Z.Y.; funding acquisition, Z.Y. All authors have read and agreed to the published version of the manuscript.

**Funding:** This work has been supported by the following funds: Hubei Provincial Science and Technology Innovative Talent Program Project, 2023DJC092. Hubei Provincial Department of Education Key Project, D20221404. Open Fund of Hunan Provincial Key Laboratory of Intelligent Electricity-Based Operation Technology and Equipment (Robot): 2022KZD1001. National Natural Science Foundation of China, 51907055. Hubei Provincial Natural Science Foundation Innovation Group Project, 2023AFA037. National Natural Science Foundation of China, 52075152. China Construction West Construction Science and Technology Research and Development Project, ZJXJZN-2022-03. China State Construction Engineering Corporation 2022 Annual Science and Technology Research and Development Project, CSCEC-2022-Q-52.

**Data Availability Statement:** No new data were created or analyzed in this study. Data sharing is not applicable to this article.

**Conflicts of Interest:** The authors declare no conflict of interest.

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
