# Peer review of "Research on Speed Control Methods and Energy-Saving for High-Voltage Transmission Line Inspection Robots along Cable Downhill"

_actuators, doi:10.3390/act12090352_

Round 1
Reviewer 1 Report
The paper presents a proposition of a cable inspection robot with energy recovery during downhill motion. The paper presents a concept of mechanical and controller electronics design. Then the design of the control algorithm is presented which uses fuzzy control to tune PID coefficients. The controller is compared in simulations with fuzzy-only and PID-only controllers. The paper considers an important problem and clearly describes the authors' approach.
My remarks and propositions of changes:
1) The title is slightly misleading, suggesting that energy saving is also a criterion for comparison of control methods, while the control methods are chosen only for speed control, and energy-saving is separate, in hardware/electronics design. I would then suggest to change the word order to "Research on Speed Control Methods and Energy-Saving for..."
Similarly, the last of the conclusions seems not justified - while it is true that the hardware setup allows energy-saving, the paper has not provided a proof that this particular method was anyhow better than any other method in terms of energy saving.
2) The two reference methods used in the experiments are not well defined: it is not described what the "fuzzy control" is (l. 443 and further) and how the parameters for the reference PID controller were chosen. Without those information the comparison of the 3 methods has no value.
3. In section 4.2, from l.388
The errors e and ec are described as deviation between desired and real (feedback) values. What are their units - are they absolute or relative values?
If they are absolute and expressed in different units - what justifies using the same ranges for membership functions?
Similar explanation should be added to output variables membership functions, as using the same functions for all outputs works well only when the the variables are of similar scale.
4) Fig. 14
The x-axis is labelled Kp/Ki/Kd, while the outputs are not those parameters, but the change of them (deltas) - please correct that as the current picture suggests using negative gains.
5) caption of Fig. 15 - I suppose it should be "fuzzy-PID controller" to distinguish from the other two methods from the experiments
Author Response
Dear Editors and reviewers:
Thank you for your valuable comments concerning our manuscript entitled “Research on Energy-Saving and Speed Control Methods for High-Voltage Transmission Line Inspection Robots Along Cable Downhill (actuators-2550570)”. The comments are all valuable and very helpful for revising and improving our paper, as well as of the important guiding significance to our researches. We have studied comments carefully and have made correction which we hope meet with approval. Revised portion are marked in red in the paper. Regarding the reviewers' feedback, the response is provided in the attached document.

Reviewer 2 Report
This paper proposes an energy saving method for control of transmission line inspection robots. The wheel motors are controlled using fuzzy PID control.
Strengths: the paper is clearly written and well-explains the problem and the proposed solution.
Weakness: Only the case of constant reference speed control is analyzed and tested.
1. The paper sets out to find energy saving control method for transmission line inspection robot. While a constant speed control problem is described and solved, how much energy saving is materialized is not clear.
2. The paper only considers downhill descent at a set of predefined descent angles, whereas for true energy efficiency, the complete line angle profile from one tower to another should be taken into account.
4. The cutoff voltage of the super-capacitor is set to half of its rated voltage (line 348). This constraint is likely active only during uphill climb.
5. The fuzzy PID approach aims to achieve a constant speed downhill motion of the inspection robot along flexible cables (line 372). Please explain how this solves the energy efficiency problem.
6. The target wheel speed for the inspection robot descending along the flexible cable is set to 6000 r/min (line 440). Please explain why the same target speed was considered for the various descent angles (10, 20, 30 deg).
7. Please comment on the energy efficiency of the fuzzy PID in terms of the various descent angles (Fig. 19).
8. The conclusion (line 478-480) does not match with the research aim described in the paper title and Abstract.
9. Sec. 5.3 considers the effects of cable wear and wind induced speed oscillations on speed control. How about including the effect of robot swinging due to cross wind?
10. Please explain how does the voltage profile of super capacitor (Fig. 28, Table 4) relate to the energy recovery? Please explain the last column in Table 4.
Author Response

(The authors gave the same response as above.)

Round 2
Reviewer 1 Report
The explanations provided address all the issues I have raised in the previous review and the modifications to the paper clarify the elements that in my opinion required clarification in the previous version of the paper.